# Label-Free Creatinine Optical Sensing Using Molecularly Imprinted Titanium Dioxide-Polycarboxylic Acid Hybrid Thin Films: A Preliminary Study for Urine Sample Analysis

**Seung-Woo Lee** [1,*], **Soad Ahmed** [1], **Tao Wang** [1], **Yeawon Park** [1], **Sota Matsuzaki** [1], **Shinichi Tatsumi** [1], **Shigekiyo Matsumoto** [2], **Sergiy Korposh** [3] **and Steve James** [4]

1   Graduate School of Environmental Engineering, The University of Kitakyushu, 1-1 Hibikino, Kitakyushu 808-0135, Japan; z8daa401@eng.kitakyu-u.ac.jp (S.A.); t-wan@kitakyu-u.ac.jp (T.W.); b0daa001@eng.kitakyu-u.ac.jp (Y.P.); souta8936@icloud.com (S.M.); s-tatsumi@kitakyu-u.ac.jp (S.T.)
2   Department of Anesthesiology, Faculty of Medicine, Oita University, 1-1 Idaigaoka, Yufu-shi, Oita 879-5593, Japan; sigekiyo@oita-u.ac.jp
3   Department of Electrical and Electronic Engineering, University of Nottingham, Nottingham NG7 2RD, UK; s.korposh@nottingham.ac.uk
4   Engineering Photonics, School of Aerospace, Transport and Manufacturing, Cranfield University, Cranfield, Bedford MK43 0AL, UK; s.w.james@cranfield.ac.uk
*   Correspondence: leesw@kitakyu-u.ac.jp; Tel.: +81-93-695-3293

**Abstract:** Creatinine (CR) is a representative metabolic byproduct of muscles, and its sensitive and selective detection has become critical in the diagnosis of kidney diseases. In this study, poly(acrylic acid) (PAA)-assisted molecularly imprinted (MI) $TiO_2$ nanothin films fabricated via liquid phase deposition (LPD) were employed for CR detection. The molecular recognition properties of the fabricated films were evaluated using fiber optic long period grating (LPG) and quartz crystal microbalance sensors. Imprinting effects were examined compared with nonimprinted (NI) pure $TiO_2$ and PAA-assisted $TiO_2$ films fabricated similarly without a template. In addition, the surface modification of the optical fiber section containing the LPG with a mesoporous base coating of silica nanoparticles, which was conducted before LPD-based $TiO_2$ film deposition, contributed to the improvement of the sensitivity of the MI LPG sensor. The sensitivity and selectivity of LPGs coated with MI films were tested using CR solutions dissolved in different pH waters and artificial urine (near pH 7). The CR binding constants of the MI and NI films, which were calculated from the Benesi–Hildebrand plots of the wavelength shifts of the second LPG band recorded in water at pH 4.6, were estimated to be 67 and 7.8 $M^{-1}$, respectively, showing an almost ninefold higher sensitivity in the MI film. The mechanism of the interaction between the template and the $TiO_2$ matrix and the film composition was investigated via ultraviolet–visible and attenuated total reflectance Fourier-transform infrared spectroscopy along with X-ray photoelectron spectroscopy analysis. In addition, morphological studies using a scanning electron microscope and atomic force microscope were conducted. The proposed system has the potential for practical use to determine CR levels in urine samples. This LPG-based label-free CR biosensor is innovative and expected to be a new tool to identify complex biomolecules in terms of its easy fabrication and simplicity in methodology.

**Keywords:** liquid phase deposition; molecular imprinting; creatinine; $TiO_2$ nanothin film; long period grating; urine analysis

## 1. Introduction

As a metabolic byproduct of muscles, creatinine (CR) is toxic for cells and is transported through the bloodstream and eliminated via renal filtration [1]. Blood and urine have different CR levels [2]; generally, the glomerular filtration rate has been universally used to diagnose and monitor kidney diseases [3,4]. For CR detection, a colorimetric method (called Jaffé reaction) [2,5] and enzymatic colorimetric methods [6,7] are commonly

used, however, such conventional methods are often interfered with by the presence of ammonium ions ($NH_4^+$) and other biological compounds such as glucose and ascorbic acid in the blood and, more significantly, in urine specimens [7]. Moreover, using a fast, accessible, cost effective, and accurate method to measure CR concentrations leads to earlier detection and better management of kidney diseases.

Label-free CR biosensing has recently attracted attention as a new noninvasive detection method [8,9]. Label-free systems can only analyze large molecules with a few readout strategies, such as when using other recognition elements like antibodies, to avoid the cost-intensive and time-consuming labeling process and challenging labeling reactions. The label-free analysis is an effective and promising strategy for fast, simple, and convenient detection of small molecules [10]. They also retain the highest degree of activity and affinity for the recognition element.

Optical fiber offers a powerful, universal platform for chemical and biological sensor fabrication through its combination with sensitive nanomaterials that exhibit specificity to a targeted chemical species [11–14]. Among various fiber optic sensing methods, long period grating (LPG) is typically produced by the periodic modulation of the refractive index (RI) of the core of an optical fiber, which enables the coupling of light between the core mode and cladding modes [15]. Consequently, this characteristic of LPG results in label-free biological and chemical sensing of target molecules, which reflect the grating resonance shift transduced by molecular reaction events that occurred on LPGs with a sensitive coating [16,17].

Recently, we demonstrated a novel optical sensor fabrication method based on the deposition of mesoporous films composed of alternate layers of silica nanoparticles ($SiO_2$ NPs) and polymers onto an LPG for the sensitive detection of organic compounds [18] and ammonia [19]. For selective optical sensing, molecular imprinting-based fiber optic LPG sensors have also been intensively developed for the nonlabeled detection of complex biomolecules [20,21]. Molecular imprinting is a promising technique for creating specific binding sites by embedding a template molecule in solid matrices. The targeted molecule can be removed from the surrounding solid matrix via appropriate chemical treatments, resulting in the formation of molecular cavities with high affinity to the template. Since the first report in 1988 [22], $TiO_2$-based molecular imprinting methods have rapidly expanded in various areas because of their several advantages, such as high affinity to organic and biological molecules, compatibility with complex components, nontoxicity, and self-cleaning effect [23,24].

In this article, we propose a CR detection method using poly(acrylic acid) (PAA)-assisted molecularly imprinted (MI) $TiO_2$ thin films. This method is based on two successive steps—surface modification of the optical fiber section containing the LPG and a base coating of $SiO_2$ NPs to make a mesoporous film. The films' porous structure resulted in the $SiO_2$ NPs coating having a high surface area but low RI. This feature of the film allowed an increase in the deposition area of $TiO_2$ deposited via liquid phase deposition (LPD). PAA is considered an additive required for controlling film growth. Thus, it was necessary to determine the optimal ratio of CR to PAA to form specific CR binding sites in the $TiO_2$ film deposited via hydrolysis of titanium hexafluoro complex precursors in water, yielding a CR@PAA/$TiO_2$ film. To investigate the imprinting effect, LPD-based $TiO_2$ films, similarly prepared by applying different ratios of CR and PAA, were also assessed. The sensitivity and selectivity of the MI LPG optical fiber sensors were tested using aqueous solutions of CR dissolved in water and an artificial urine matrix with a composition similar to that of real human urine. To the best of our knowledge, this is the first trial for CR molecular imprinting using $TiO_2$ nanothin films coupled with fiber-optic LPGs to accomplish an amplified RI sensor response.

## 2. Materials and Methods

### 2.1. Materials

Ammonium hexafluorotitanate (IV) ($[NH_4]_2TiF_6$, Mw: 197.93) was purchased from Morita Chemical Industries, Japan. $H_3BO_3$, HCl (1.0 M), and $H_2O_2$ were purchased from Wako Pure Chemicals, Japan. Poly(diallyldimethyl ammonium chloride) (PDDA: Mw = 200,000–350,000 g mol$^{-1}$, 20 wt% in water), PAA (Mw = ca.100,000 g mol$^{-1}$, 35 wt% solution in water), and $H_2SO_4$ (98%) were purchased from Sigma-Aldrich, USA. A colloidal solution of $SiO_2$ NPs (SNOWTEX 20L, 40–50 nm) was purchased from Nissan Chemical, Japan. The ingredients of the artificial urine listed in Table 1 were chosen and mixed following previously reported recipes [25,26]. $CaCl_2$, $Na_2SO_4$, NaCl, glucose, spermine, caffeine, and *L*-amino acids—histidine, glycine, threonine, tyrosine, isoleucine, alanine, arginine, proline, glutamic acid, valine, aspartic acid, and lysine—were purchased from Wako Pure Chemicals, Japan. $NaH_2PO_4 \bullet 2H_2O$, KCl, KOH, and urea were purchased from Kanto Chemical, Japan. $NH_4Cl$ was purchased from Nacalai Tesque, Japan. $Na_2SO_4$, NaCl, uric acid, glucose, and sarcosine were purchased from Tokyo Kasei, Japan. Other chemicals used as solvents were of analytical grade purity and obtained from commercial sources. All chemicals were guaranteed reagents and used without further purification. Deionized (DI) water (18.2 M$\Omega \cdot$cm) was obtained by reverse osmosis, followed by ion exchange and filtration (Aquapuri 541; Young In Scientific, Seoul, Korea). An LPG of 35 mm length and 100-μm period was fabricated point-by-point in a single-mode optical fiber (PS750; Fibercore, Southampton, UK) by exposing the optical fiber to the output from a frequency quadrupled Nd:YAG laser operating at 266 nm.

**Table 1.** Composition of the artificial urine medium [25,26].

| Component | Conc. (mmol L$^{-1}$) | Component | Conc. (mmol L$^{-1}$) | Component | Conc. (mmol L$^{-1}$) |
|---|---|---|---|---|---|
| $CaCl_2$ | 4 | Glucose | 2 | Isoleucine | 5 |
| $NaH_2PO_4 \cdot 2H_2O$ | 14 | Spermine | 0.01 | Alanine | 5 |
| KCl | 40 | Sarcosine | 0.02 | Arginine | 5 |
| $NH_4Cl$ | 85 | Histidine | 5 | Proline | 5 |
| $Na_2SO_4$ | 15 | Glycine | 5 | Glutamic acid | 5 |
| NaCl | 60 | Threonine | 5 | Valine | 5 |
| Urea | 42 | Tyrosine | 5 | Lysine | 5 |

### 2.2. Preparation of a Mesoporous Base Coating on LPG

Before the deposition of the $TiO_2$ coating via LPD, the region of the optical fiber containing the LPGs was rinsed with DI water and immersed in 1 wt% ethanolic KOH (ethanol:water, 3:2, *v/v*) solution for 20 min, affording a negatively charged surface. Afterward, the optical fiber was modified via alternate layer-by-layer (LbL) deposition of PDDA and $SiO_2$ NPs to provide a mesoporous base coating. The detailed procedure of the deposition of $SiO_2$ NPs onto the LPGs has been previously reported [27]. Briefly, after KOH treatment to cover the surface of the optical fiber cladding with OH groups, the fiber was alternately immersed in a 0.5 wt% solution containing a positively charged polymer, PDDA, and in a colloidal solution containing negatively charged $SiO_2$ NPs after drying and washing, each for 15 min (Scheme 1). The fiber was rinsed with DI water and dried by flushing with $N_2$ gas after each deposition step. This process was repeated until the required coating thickness was achieved. In the following text, this base coating is expressed as (PDDA/$SiO_2$ NPs)$_n$, where n is the number of deposited bilayers and the outermost layer of the base coating was covered with positively charged PDDA. After the deposition of a base coating, the optical fiber was coated with either a CR-MI or nonimprinted (NI) $TiO_2$ nanothin film.

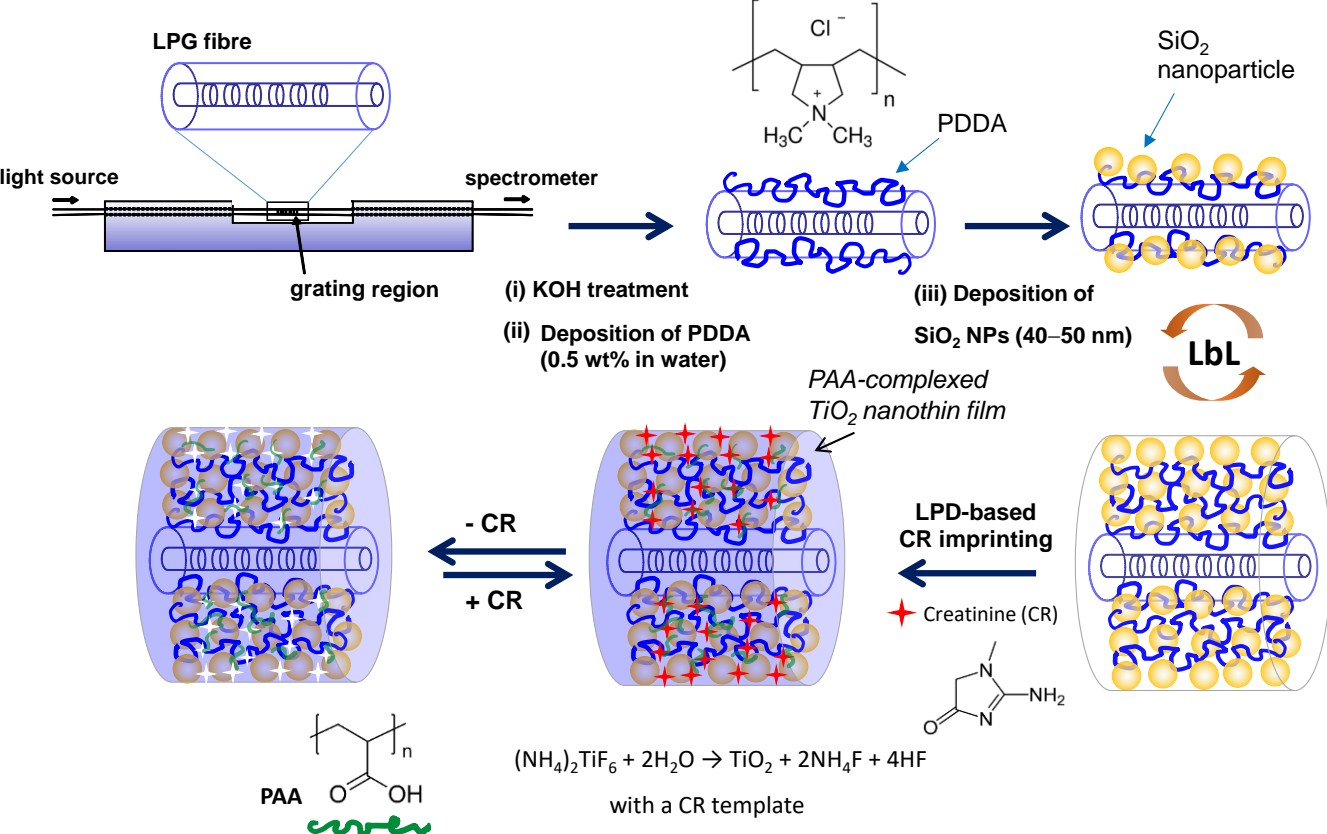

**Scheme 1.** Schematic of the LPD-based CR imprinting using an LPG optical fiber coated with a mesoporous LbL base coating comprising PDDA and SiO$_2$ NPs.

Transmission spectra (TS) of LPGs were acquired by passing the output of a tungsten-halogen lamp (HL-2000; Ocean Optics, FL, USA) into the optical fiber and analyzing the transmitted light using a fiber-coupled charge-coupled device spectrometer (HR-2000; Ocean Optics, FL, USA).

*2.3. LPD-Based CR Imprinting on LPGs*

To prepare an MI film for CR detection, first, PAA and CR were mixed at various compositions and then mixed with a TiO$_2$ film-forming solution of a mixture of (NH$_4$)$_2$TiF$_6$ and H$_3$BO$_3$. Finally, the solution was infused to a PDDA/SiO$_2$ NP LbL base coating deposited on the optical fiber for subsequent coating with a TiO$_2$ film via LPD for CR imprinting (Scheme 1). Briefly, a mixture of CR and PAA in water was mixed with an aqueous solution containing (NH$_4$)$_2$TiF$_6$ (100 mM) and H$_3$BO$_3$ (500 mM) at a 4.5:4.5:1 (*v/v*) ratio of (NH$_4$)$_2$TiF$_6$:H$_3$BO$_3$:CR/PAA, where the final concentrations of CR and PAA in the film-forming solution for CR imprinting were adjusted in the range of 0.1–5 mM and 0.1–6 mM (as monomer unit concentration), respectively. A similar film-forming solution without CR or both CR and PAA was used for NI film fabrication. The coating solution (350 µL) was placed into a Teflon deposition cell containing a recess (0.5 cm wide, 6.0 cm long, and 0.3 cm deep) for 3 h. Then, the coating solution was removed by suction, and the LPG optical fiber was washed with DI water, followed by drying with N$_2$ gas. After the film deposition, LPGs modified with the LPD-based TiO$_2$ films were kept in an oven at 60 °C and humidity 90% for 12 h to improve TiO$_2$ crystallinity.

To assess the sensitivity to LPGs, the prepared films were washed with an HCl solution (pH 3.5) for 15 min to remove the CR template. For template rebinding and guest selectivity tests, the CR-MI- and NI-LPG sensors were exposed to different concentrations of CR dissolved in water and artificial urine for 5 min and to individual amino acids as guest molecules at 4 mM in water.

### 2.4. Film Characterization

Ultraviolet–visible (UV–Vis) absorption spectra of the films assembled on quartz substrates were measured using a JASCO V-570 UV–Vis spectrophotometer. Before the film deposition, a quartz plate was cleaned with concentrated sulfuric acid (96%), rinsed several times with DI water, treated with 1 wt% KOH in a mixture of ethanol and water (3:2, *v/v*) for 20 min with sonication, rinsed with DI water, and dried with $N_2$ gas.

Gold-coated quartz crystal microbalance (QCM: Nihon Dempa Kogyo Co., Ltd., Tyoko, Japan) electrodes with a 9 MHz fundamental oscillation frequency and a ca. 0.196 cm$^2$ active surface area of one side were used to track a mesoporous base coating deposition. Before the film deposition, electrodes were air–plasma-treated using a plasma generator (Covance, Femto Science Inc., Hwaseong, Korea), washed with water and ethanol, and then dried using $N_2$ gas. Separately, the CR rebinding on the CR-MI and NI films was confirmed using a 30 MHz QCM twin electrode with two channels (Nihon Dempa Kogyo Co., Ltd., Tyoko, Japan). The MI or NI film deposition was conducted on each electrode channel, following the procedures described in Section 2.3; one is for a working electrode with a CR@PAA/TiO$_2$ film, and the other is for a reference electrode with a PAA/TiO$_2$ film.

Field-emission scanning electron microscope (FE-SEM) measurements were performed using a S-5200 apparatus (Hitachi, Ltd., Tokyo, Japan) for the substrates with a (PDDA/SiO$_2$ NPs)$_{10}$ base coating before and after CR imprinting, similar to that used to fabricate the LPG sensor. To investigate the composition of the LPD-based films deposited on LPGs, solid samples collected from the LPD film-forming solutions after film deposition were analyzed using an FT/IR-4600 (JASCO Corp., Tokyo, Japan) attenuated total reflectance Fourier-transform infrared spectrophotometer (ATR-FTIR). X-ray photoelectron spectroscopy (XPS) measurements of the collected solid samples were performed on AXIS-His (Shimazu-Kratos, Co., Ltd., Kanagawa, Japan) using Mg Kα (1253.6 eV) radiation. The applied power was operated at 15 kV and 10 μA. The base pressure in the analysis chamber was less than $10^{-8}$ Pa. All peaks were calibrated with respect to the C 1s peak at 285.0 eV as the reference. The surface morphology of the (PDDA/SiO$_2$ NPs)$_{10}$ base coating was also studied using a JSPM-5200 (JEOL Ltd., Tokyo, Japan) atomic force microscope (AFM) working in a noncontact mode using a MicroMash NSC12/Ti-Pt/15 silicon cantilever (<40 nm curvature tip radius and 15–20 μm tip length).

## 3. Results and Discussion

### 3.1. LbL Base Coating of PDDA and SiO$_2$ NPs on LPG

Figure 1a shows the changes in the TS of the 100 μm period LPG due to the alternate deposition of PDDA and SiO$_2$ NPs. The spectra were recorded in the silica colloidal solution after each deposition of SiO$_2$ NPs on the PDDA layer reached saturation since the change in the TS was more significant when the outermost layer consisted of SiO$_2$ NPs. During the alternate PDDA and SiO$_2$ NP deposition, the LPG TS undergoes a dramatic change: two resonance bands at 705 (first) and 953 (second) nm shifted in opposite directions (Figure 1b,c). The observed response of the LPG TS to increasing optical thickness (the product of the RI and geometrical thickness) agreed with previous reports [18,19,27]. The observation of a linear wavelength shift with the increase in deposition cycles demonstrated that the film was uniformly deposited (Figure 1d). Average wavelength shifts per deposited layer were estimated to be 0.5 and 1.3 nm for the first and second resonance bands, respectively.

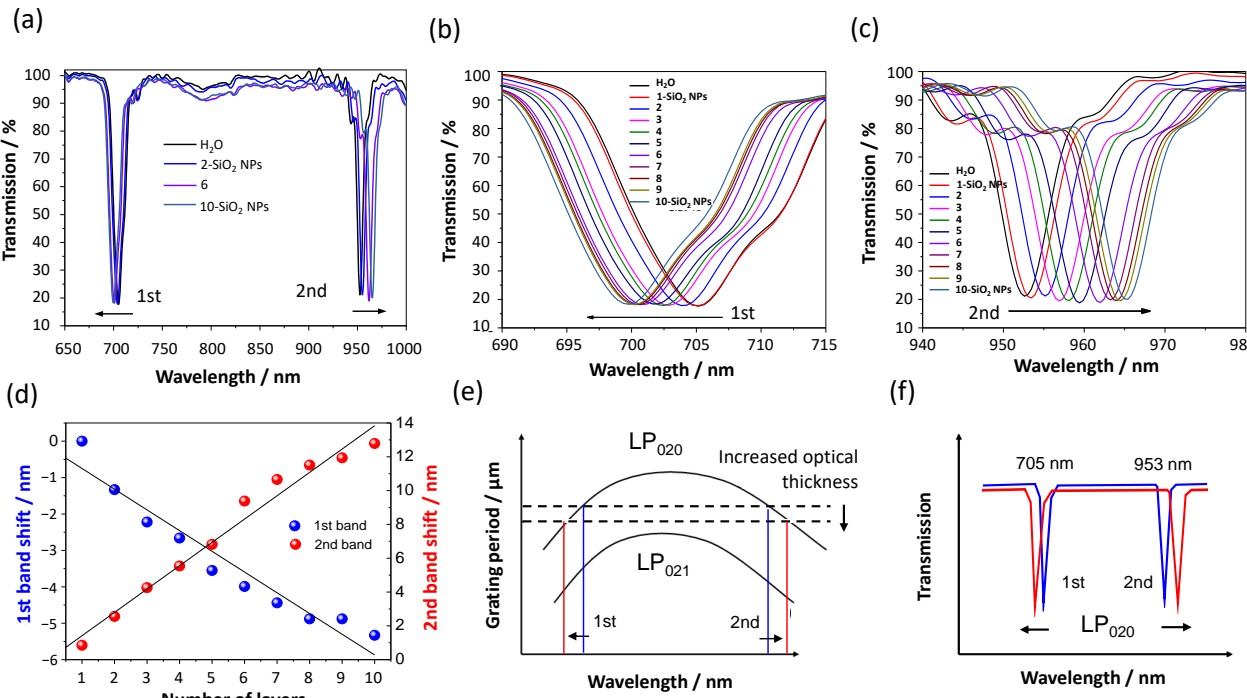

**Figure 1.** (**a**) Changes in LPG TS due to the deposition of PDDA and $SiO_2$ NPs. (**b**,**c**) Enlarged TS of the first and second resonance bands, respectively. (**d**) Dependence of wavelength shifts of the first and second resonance bands upon $SiO_2$ NPs deposition cycles. (**e**) Illustration of the phase-matching properties of an LPG of length 35 mm with a period of 100 μm. (**f**) Schematic of wavelength shifts of the $LP_{020}$ resonance bands at 705 and 953 nm due to the increased optical thickness.

The observed changes in LPG TS could be explained by considering the operation of LPG. LPG acts to couple light from the forward propagating mode of the core of the fiber to a discrete set of copropagating cladding modes at wavelengths governed by the phase-matching condition:

$$\lambda_{(x)} = (n_{core} - n_{clad(x)})\Lambda, \quad (1)$$

where $\lambda_{(x)}$ represents the wavelength at which coupling occurs to the linear polarized ($LP_{0x}$) mode, $n_{core}$ is the effective RI of the mode propagating in the core of the fiber, $n_{clad(x)}$ is the effective RI of the $LP_{0x}$ cladding mode, and $\Lambda$ is the period of LPG. The behavior of the TS of an LPG with a 100 μm period has been well described [18,27,28], where the LPG is shown to be sensitive to changes in the optical thickness of coatings deposited onto the surface of the optical fiber cladding. For the 100 μm period LPG used in this study, when combined with the 10-cycle deposition of an alternate layer of PDDA and $SiO_2$ NPs, it allowed phase-matching to the $LP_{020}$ cladding mode (Figure 1e), with a resulting LPG transmission spectrum of the form shown in Figure 1f. Therefore, resonance bands coupled to the $LP_{020}$ cladding mode showed a tendency to spread to the left and right at around 705 and 953 nm.

Figure 2a,b show the surface morphology and cross-section, respectively, of a $(PDDA/SiO_2 \text{ NPs})_{10}$ film deposited on a silicon wafer. The $(PDDA/SiO_2 \text{ NPs})_{10}$ film has a uniform surface consisting of $SiO_2$ NPs with an average 45 nm diameter (Figure 2a). The film thicknesses obtained after the deposition of the $(PDDA/SiO_2 \text{ NPs})_{10}$ film, determined from the FE-SEM cross-section measurements, were approximately 320 nm (Figure 2b). This film thickness corresponded to approximately 77% of a theoretical value (416 nm) estimated from the hexagonal close-packing of $SiO_2$ NPs with an average 45 nm diameter under the assumption that the distance separation between the particles adsorbed per layer was approximately 40 nm. In our previous study [27], the pore size distribution of PDDA/$SiO_2$ NP LbL film was confirmed, showing a well-developed mesoporous structure with a 12.5 nm average pore radius and 50 $m^2$ $g^{-1}$ specific surface area. Figure 2c,d show

AFM images of the surface morphology of the (PDDA/SiO$_2$ NPs)$_{10}$ film, which agreed with the results obtained from SEM measurements. The roughness of the (PDDA/SiO$_2$ NPs)$_{10}$ film was estimated to be 12.3 nm, which is close to 19.7 nm, indicating a difference in the height of the outermost layer of the 77% close-packed SiO$_2$ NP film.

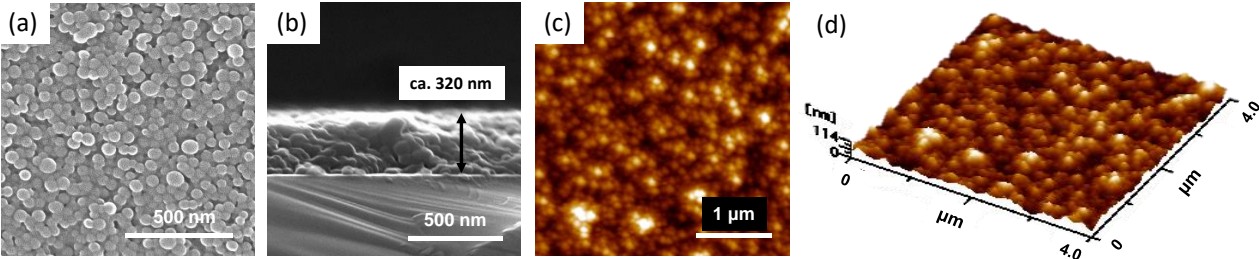

**Figure 2.** FE-SEM images of (**a**) surface morphology and (**b**) cross-section of the base coating of a (PDDA/SiO$_2$ NPs)$_{10}$ film deposited on a silicon wafer. (**c**) 2D and (**d**) 3D AFM images of the surface morphology of the corresponding film.

The uniform deposition of the base coating was also confirmed using 9 MHz QCM frequency shifts measured during the same LbL electrostatic self-assembly, showing a linear decrease with an average frequency shift of 28 ± 17 and 1430 ± 115 Hz per cycle of PDDA and SiO$_2$ NPs, respectively (Figure S1). These QCM results agreed with the wavelength shifts of the two LPG resonance bands, which were much developed after SiO$_2$ NP deposition. The total frequency shift during 10 cycle deposition was approximately 14,600 Hz, which corresponded to a mass increase of 16206 ng when a frequency decrease of 1 Hz was assumed to be 1.1 ng in the current system. The density ($\rho$) of adsorbed materials on one side of the QCM electrode can be estimated using the following modified equation [29]:

$$\rho = 0.027\,(-\Delta F\,(\text{Hz}))/d, \tag{2}$$

where $d$ is the thickness of adsorbed layers. From the SEM film thickness (ca. 320 nm) and QCM frequency (14,600 Hz) results, the film density of the base coating was estimated to be approximately 1.3 g cm$^{-3}$—between those of PDDA (1.2 g cm$^{-3}$) [29] and SiO$_2$ NPs (approximately 2 g cm$^{-3}$) [30]. Surprisingly, the film density reached approximately 78% of the average density (1.6 g cm$^{-3}$) of both components, which was almost the same as the packed degree of SiO$_2$ NPs in the film (77%), observed in the SEM cross-section view.

*3.2. Optimization of TiO$_2$ Film Matrix for CR Imprinting*

Figure 3a shows the normalized UV–Vis absorption spectra of the LPD-based films with different amounts of PAA and CR employed in the TiO$_2$ matrix measured to confirm the optimized film composition for film coating and CR imprinting. The pure TiO$_2$ film without CR and PAA showed the largest absorbance at 236 nm, attributable to the absorption band of TiO$_2$, indicating the relatively fast and thick TiO$_2$ film growth on the quartz substrate. Using the LPD coating solution containing 2 mM CR, the absorption band of TiO$_2$ at 236 nm slightly decreased and showed a red shift of ~6 nm. However, the absorption in the visible light region of 350–500 nm increased due to the presence of CR (inset of Figure 3a), which was probably due to the interaction between the template molecule and the TiO$_2$. Meanwhile, the TiO$_2$ absorption band at around 236 nm was significantly decreased by incorporating PAA into the LPD film-forming solution at 6 mM monomer unit molar concentration, which suggested that the LPD-based film formation could be suppressed by the complexation of TiO$_2$ with PAA.

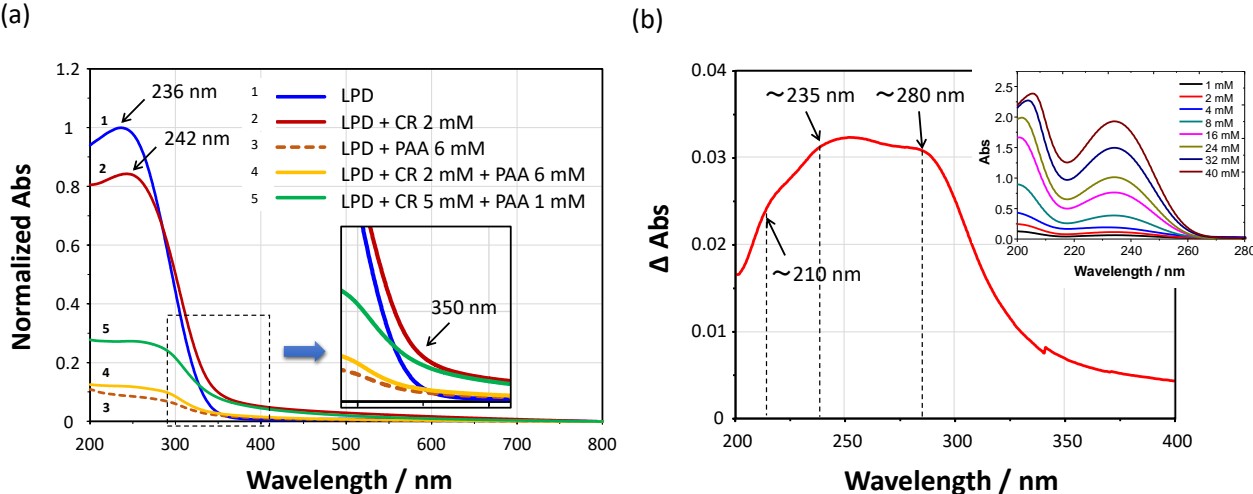

**Figure 3.** (**a**) Normalized UV–Vis absorption spectra of the LPD-based films with different amounts of PAA and CR employed in the TiO$_2$ matrix deposited on a quartz substrate. (**b**) Difference spectrum obtained from the normalized UV–Vis absorption spectra of the two PAA-assisted TiO$_2$ films fabricated with and without CR (2 mM CR and 6 mM PAA). The inset shows UV–Vis absorption spectra corresponding to the different concentrations of CR in water.

Remarkably, when CR was used simultaneously with PAA (2:6, mM/mM) for film coating, the absorption band in the ultraviolet (UV) range slightly increased. The difference between the normalized UV–Vis absorption spectra of the two PAA-assisted TiO$_2$ films (6 mM) with and without CR (2 mM) is shown in Figure 3b. Three characteristic absorption bands at around 210, 235, and 280 nm were observed. The first two absorption bands reflected the absorption peaks of CR in the UV range (inset of Figure 3b), whereas the third band might be due to the protonated amino functional groups (-N$^+$-H and =N$^+$-H), which could make complexes with anionic moieties derived from the functional groups of TiOH and -COOH-TiO$^-$ and -COO$^-$. From the above results, we inferred that the TiO$_2$ film growth via LPD was highly affected by the presence of PAA, which could be explained by noting that PAA formed a complex with TiF$_6$$^{2-}$ in the film-forming solution and inhibited the nuclear formation of TiO$_2$. Moreover, decreasing the concentration of PAA to 1 mM enhanced TiO$_2$ film formation. In addition, the absorption in the visible range of 350−500 nm largely increased when a higher concentration of CR (5 mM) was applied, similar to that observed for the TiO$_2$ film prepared using the film-forming solution containing 2 mM CR, which suggested that increasing the relative concentration of CR enhances binding site formation for CR in TiO$_2$ films.

### 3.3. Confirmation of Optimal CR Imprinting Using QCMs

The CR rebinding into the CR-MI TiO$_2$ films was confirmed using a 30 MHz twin electrode with the aid of a NAPICOS analyzer system (Nihon Dempa Kogyo Co., Ltd., Japan), which was self-calibrated to the external condition changes by simultaneous measurements of the control and data signals. The QCM sensor was fixed in the thermostatic flow cell; one channel was modified with a pure TiO$_2$ or PAA/TiO$_2$, whereas the other was modified with a CR@PAA/TiO$_2$ film. After HCl treatment (pH 3.5) for template removal, the sensor response on each channel was confirmed by flowing the CR solutions of different concentrations of 0–40 mM in water (pH 4.6) with an intermediate HCl washing after CR rebinding test at each concentration. The frequency shifts of both channels due to the LPD-based film deposition were adjusted to be approximately 20,000 Hz measured in the air after thorough washing with DI water and drying with N$_2$ gas. According to the Sauerbrey equation [31], a frequency decrease of 1 Hz corresponded to a mass of 0.04 ng in the system.

Figure 4a shows a comparison of dynamic QCM responses due to the successive in situ CR rebinding of the pure TiO$_2$ (control) and CR@PAA/TiO$_2$ (0.1 mM CR and 0.1 mM PAA)

films coupled with the twin electrode. Moreover, the CR binding was largely suppressed in the PAA/TiO$_2$ (control and 1 mM PAA; hereinafter, NI) film, whereas the CR@PAA/TiO$_2$ (3 mM CR and 1 mM PAA; hereinafter, CR-MI) film showed relatively large responses to the CR (Figure 4b). Notably, the MI film showed a much faster recovery by HCl washing than the NI film, although the baseline was not completely recovered in both films.

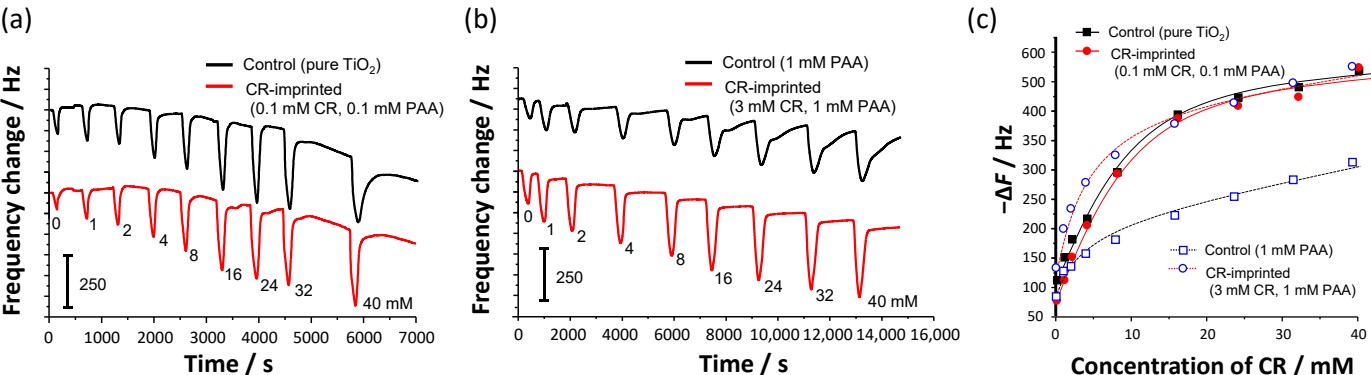

**Figure 4.** Dynamic QCM responses due to the successive in situ CR rebinding of (**a**) pure TiO$_2$ (control) and CR@PAA/TiO$_2$ (0.1 mM CR and 0.1 mM PAA) films and (**b**) coupled PAA/TiO$_2$ (control and 1 mM PAA) and CR@PAA/TiO$_2$ (3-mM CR and 1-mM PAA) films coupled with a twin electrode and used after HCl washing. (**c**) The corresponding frequency shifts due to the CR rebinding for the four LPD-based MI and NI TiO$_2$ films.

No significant difference in both electrodes in Figure 4a was observed in the given concentration range of CR (Figure 4c), indicating that the amounts of PAA and CR employed in the film were too small to create specific binding sites for CR. Remarkably, the responses of the MI film in a lower concentration range (Figure 4c), less than 10 mM CR, were larger than those of the film prepared in low concentrations of CR and PAA (0.1-mM CR and 0.1 mM PAA). However, both films exhibited similar adsorption isotherms, approaching near saturation with a frequency change of approximately 500 Hz at CR concentrations of more than 30 mM. Particularly, a longer response time in the MI film, which took approximately two times as long as the CR@PAA/TiO$_2$ (0.1 mM CR and 0.1 mM PAA) to reach saturation, indicated the formation of CR binding sites inside the film.

From these QCM results, we inferred that the imprinted structure for effective CR rebinding could be optimized by the amounts of CR and PAA employed in the TiO$_2$ matrix, where the driving force of CR binding seemed to depend on the electrostatic interaction with TiO$_2$ rather than the acid–base reaction with PAA. However, nonspecific binding of CR in the MI film might be suppressed by the incorporation of PAA, which helped control the activity of TiOH residues in the binding cavity and improved the stability of the sensor response. We also considered further increasing the ratio of CR to PAA to maximize the film-imprinting effect.

### 3.4. Evaluation of CR Imprinting on LPG

Figure 5 shows the TS of the LPG coated with a (PDDA/SiO$_2$ NPs)$_{10}$ base coating measured in water before and after CR@PAA/TiO$_2$ deposition (5 mM CR and 1 mM PAA) film for 3 h, after heat treatment at 60 °C for 12 h, and after template removal. As seen from the enlarged views of Figure 5b,c, the two LPG resonance bands had central wavelengths around 692 and 980 nm after 3 h deposition, showing blue and red wavelength shifts of ca. 7 and ca. 14 nm, respectively, relative to the wavelengths at the start of the process (699 and 966 nm, respectively). During LPD film deposition, the transmission at wavelengths between the two resonance bands showed no appreciable change, indicating that the optical thickness of the CR@PAA/TiO$_2$ film could not reach the phase-matching turning point of the LP$_{21}$ mode (Figure 1e), which would have been characterized by the development of a broad attenuation band centered at around 825 nm.

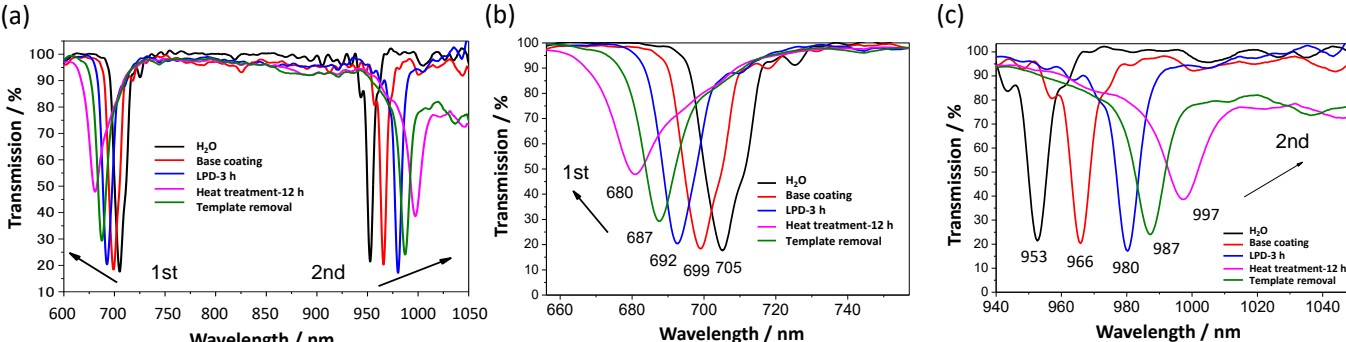

**Figure 5.** (**a**) Changes in the TS of the LPG coated with a base coating of (PDDA/SiO$_2$ NPs)$_{10}$ measured in water before and after CR@PAA/TiO$_2$ (5 mM CR and 1 mM PAA) film deposition for 3 h, after heat treatment at 60 °C for 12 h, and after template removal. Enlarged views of the changes in the TS of the (**b**) first and (**c**) second resonance bands due to the corresponding procedures for CR imprinting.

Further, a blue shift of the first resonance band at around 692 nm and a red shift of the second resonance band at around 980 nm were observed following heat treatment, i.e., 692–680 and 980–997 nm for the first and second resonance bands, respectively, which meant that the crystallinity of TiO$_2$ increased upon heat treatment, thereby increasing the RI of the CR@PAA/TiO$_2$ film. All results relevant to the wavelength shifts of both resonance bands are summarized in Table 2. Moreover, a similarly prepared PAA/TiO$_2$ (1 mM PAA) film without CR exhibited different optical features during the imprinting process (Figure S2). The wavelength shifts of the two resonance bands were relatively small compared with those of the CR@PAA/TiO$_2$ (5 mM CR and 1 mM PAA) film (Table 2).

**Table 2.** Wavelength shifts of the resonance bands after the successive procedures of LPD-based TiO$_2$ coating, heat treatment, and HCl washing for CR imprinting.

| Film Name (CR:PAA, mM/mM) | Wavelength Shift (nm) | | | | | |
| --- | --- | --- | --- | --- | --- | --- |
| | LPD Coating | | Heat Treatment [a] | | HCl Washing [b] | |
| | First | Second | First | Second | First | Second |
| CR@PAA/TiO$_2$ (5:1) | −7.0 | 14 | −12 | 17 | 7.0 | −10 |
| CR@PAA/TiO$_2$ (0:1) | −4.6 | 10 | −8.3 | 13 | 0.5 | −0.9 |
| CR@PAA/TiO$_2$ (2:6) | −8.0 | 6.0 | 0 | 6.0 | 0 | 0 |

[a] At 60 °C (humidity: 90%) for 12 h; [b] at pH 3.5 for 15 min.

To confirm the effects of the base coating and the PAA concentration on the LPG band shift, a CR@PAA/TiO$_2$ (2 mM CR and 6 mM PAA) film without a base coating was fabricated similarly. As shown in Figure S3, the film exhibited relatively small wavelength shifts for both resonance bands. After heat treatment, the first and second resonance bands, which were found originally at around 705 and 953 nm, shifted in opposite directions to 697 and 965 nm, respectively. As summarized in Table 2, the band separation of the second resonance band before and after heat treatment decreased in the order of CR@PAA/TiO$_2$ (5 mM CR and 1 mM PAA) > PAA/TiO$_2$ (1 mM PAA) > CR@PAA/TiO$_2$ (2 mM CR and 6 mM PAA), showing 31, 23, and 12 nm, respectively. This larger band separation in the films with a base coating might be due to the high surface area of the base coating, thereby increasing the RI due to the enhanced amount of TiO$_2$ deposited between the SiO$_2$ NPs [32].

To remove the CR template embedded in the TiO$_2$ matrix, the CR@PAA/TiO$_2$ films were treated with an aqueous solution of HCl (pH 3.5) for 15 min. As a result, wavelength shifts of the first and second resonance bands from 680 and 997 nm to 687 and 987 nm, respectively, were observed in the CR@PAA/TiO$_2$ (5 mM CR and 1 mM PAA) film, suggesting a reduction in the RI of the coating (Figure 5b,c), which indicated that the CR template molecules could be removed from the film; we defined it as the MI film. Remarkably,

minute wavelength shifts less than 1 nm were observed for both resonance bands after HCl washing (Table 2), suggesting the absence of the template in the NI film. Oddly, no wavelength shift of both LPG bands after HCl washing was observed in the CR@PAA/TiO$_2$ (2 mM CR and 6 mM PAA) film (Figure S3), despite the existence of CR in the film-forming solution, indicating that the higher the PAA content, the smaller the amount of CR introduced into the film. From the above experimental results based on UV–Vis, QCM, and LPG measurements, the optimal concentration of CR should be considered in the range of 3–5 mM, along with the incorporation of 1 mM PAA in the film-forming solution.

### 3.5. Sensitivity and Selectivity of the CR-MI LPG Sensor

The prior treatment of the LPG in HCl solution at pH 3.5 induced a blue shift of approximately 10 nm of the second resonance band in the MI film indicating that the TiO$_2$ and PAA deposited via LPD contributed to a 21 nm wavelength shift for the second LPG band, approximately 68% of the total 31 nm wavelength shift observed after heat treatment. Remarkably, the NI film showed a similar wavelength shift of approximately 22 nm for the second LPG band after HCl washing, indicating a film composition similar to that of the MI film.

The MI film exhibited meaningful wavelength shifts of the LPG resonance bands upon rebinding the template into the matrix. Figure 6a,b show the responses of the MI film-modified LPG sensor to different concentrations of CR in water (pH 6.8). The wavelength shifts of the resonance bands reached near saturation as the concentration of CR increased to 25 mM, showing blue and red shifts of ~1.8 and ~3.4 nm, respectively, for the first and second resonance bands upon 25 mM CR rebinding (Figure 6c). The second resonance band showed about two times the sensitivity to CR compared with that of the first resonance band. Remarkably, the HCl-treated CR@PAA/TiO$_2$ (2 mM CR and 6 mM PAA) film showed no response upon exposure to 20 mM CR (Figure S3), indicating the absence of useful binding sites for CR detection. The formation of CR binding sites in the LPD-based TiO$_2$ film could be determined by the subtle difference in the ratio of CR and PAA. Consequently, the MI film could be successfully fabricated with a ratio of 5:1 for CR and PAA, and MI sites for CR rebinding were created in the film.

$$G + BS \overset{K_a}{\Leftrightarrow} G/BS \tag{3}$$

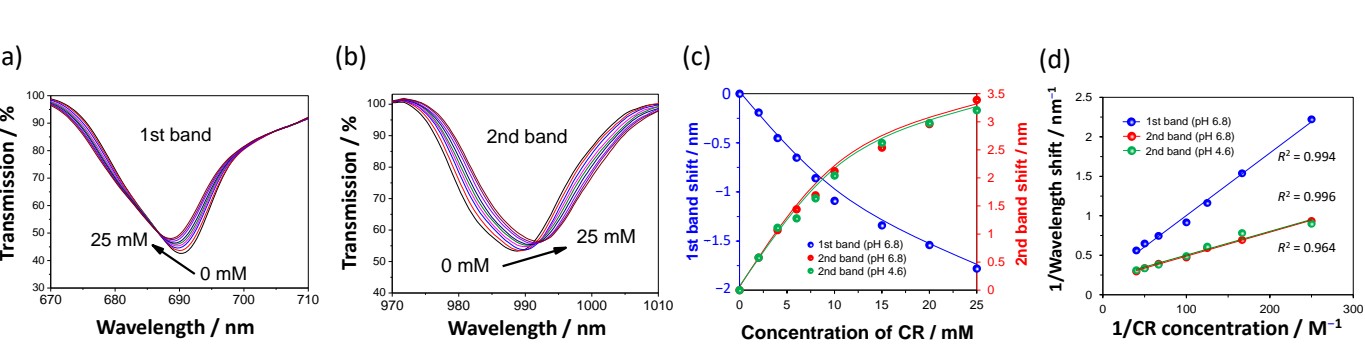

**Figure 6.** Enlarged views of the TS of the (**a**) first and (**b**) second resonance bands of the MI film-modified LPG upon rebinding of different concentrations of CR in water (pH 6.8). (**c**) Calibration curves of the wavelength shifts of the first (at pH 6.8) and second (at pH 4.6 and 6.8) resonance bands. (**d**) Benesi–Hildebrand plots obtained from the wavelength shifts of the LPG resonance bands for CR rebinding under different pH conditions.

As evident in Figure 6c, the adsorption of CR in the MI film could be described using Langmuir adsorption isotherms of the first and second resonance bands. If a 1:1 stoichiometry of the complex is assumed, the binding constant, $K_a$, between the guest (G) and binding site (BS) in the MI film can be defined by Equation (3), which is further simplified using the Benesi–Hildebrand method [33]. Benesi–Hildebrand plots for the CR template rebinding, which were based on the wavelength shifts of the first and second resonance bands recorded in water at pH 6.8, yielded binding constants of 27 ($R^2$ = 0.994)

and 62 M$^{-1}$ ($R^2 = 0.996$), respectively (Figure 6d), showing an approximately twofold higher sensitivity of the second resonance band. Consequently, the corresponding dissociation constant, $K_d$, was estimated to be 0.037 and 0.016 M for the first and second resonance bands, respectively. Remarkably, the CR binding of the MI film was almost unaffected by the pH of the solution, showing $K_a$ and $K_d$ values of 67 M$^{-1}$ and 0.015 M ($R^2 = 0.964$), respectively, for the second resonance band recorded in water at pH 4.6, indicating that the assessment of CR present in aqueous solutions was possible under variable pH conditions, suggesting a high potential of real urine sample analysis.

Figure 7a compares the calibration curves of the wavelength shifts of the second LPG band upon exposure of MI and NI films to different concentrations of CR at pH 4.6. The CR adsorption isotherms of the MI and NI films were completely different, showing a relatively high sensitivity to CR in the MI film. The low sensitivity of the NI film to CR meant that the incorporation of PAA into the film suppressed the nonspecific binding of CR onto TiO$_2$ matrices, which was expected from the QCM results (Figure 4).

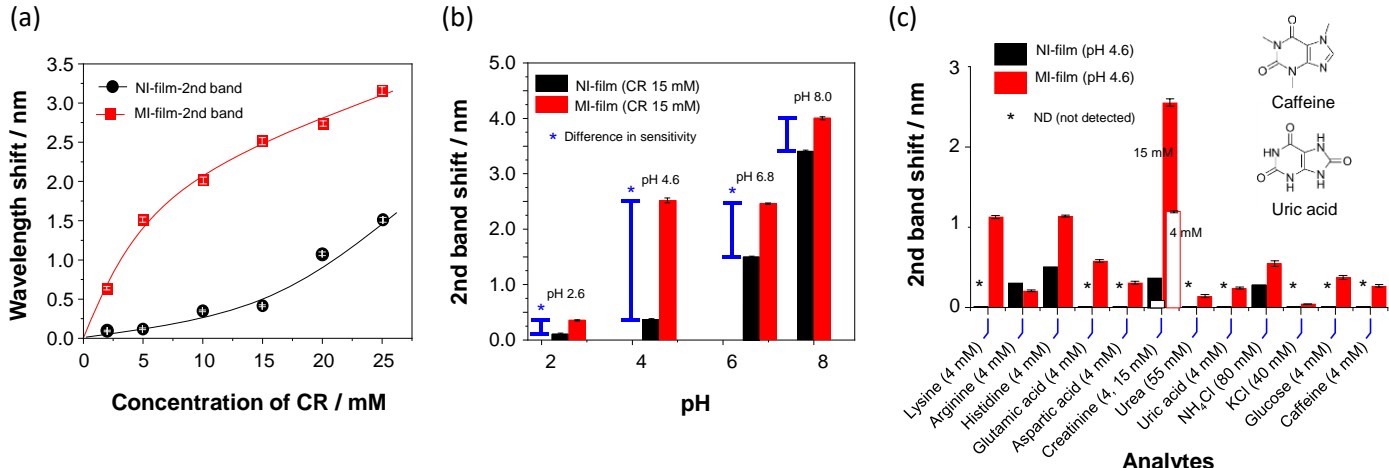

**Figure 7.** (**a**) Calibration curves of the wavelength shifts ($n = 3$) of the second LPG band upon exposure of the MI and NI films to different concentrations of CR at pH 4.6. (**b**) Comparison of the sensitivity of the MI and NI films to CR (15 mM) based on the wavelength shifts ($n = 3$) of the second LPG band under four pH conditions. (**c**) Comparison of the wavelength shifts ($n = 3$) of the second LPD band upon exposure of the NI and MI films to the template (4 and 15 mM) and guest molecules (4 mM for each organic compound).

As mentioned above, the MI film revealed almost the same responses when exposed to different concentrations of CR at pH 4.6 and 6.8. Meanwhile, the NI film showed relatively enhanced sensitivity to CR at pH 6.8 compared with the MI and NI films for CR at 15 mM (Figure 7b). Presumably, this reflected increased nonspecific BSs in the NI film with the increase in pH due to the dissociation of the T$^{i4+}$-PAA complexes present near the TiO$_2$ surface, as shown in the equilibrium reaction (Equation (4)). As reported in the literature [34,35], the p$K_a$ of the PAA is 4.5, so that the carboxylic acid groups in the polymer will exist as complexed with the TiO$_2$ at pH <4.5 and as a half mixture of both -COOH and -COO$^-$ groups at pH 4.5. At pH 6, polyelectrolyte dissociation increased rapidly, showing almost no -COO$^-$ groups in the polymer chain [36]. The NI film being more pH-dependent than the MI film suggested that the guest binding in the MI film did not depend on the nonspecific binding to the carboxylate groups.

$$\text{T}^{i4+}\text{-O-C(O)-PAA} + \text{H}_2\text{O at pH 4.6} \leftrightarrows \text{T}^{i4+}\text{-O}^- + {}^-\text{O-C(O)-PAA} + 2\text{H}^+ \tag{4}$$

Apparently, at pH < 4.6 or >6.8, both MI and NI films showed extremely different sensor responses to CR. In addition, CR binding was very ineffective at pH 2.6 for both films. However, the sensitivity of both films significantly increases at pH 8, but the difference between both films decreased, which implied that the nonspecific binding in the NI film

was much significant, along with the pH increase. Based on these results, the optimal pH was considered to be 4.6; however, a neutral condition (pH 6.8) might also be acceptable for neutral real urine sample analysis because the MI film was almost not interfered with in its sensitivity.

To confirm the best selectivity of the MI film, the CR template and other guest molecules (each 4 mM) dissolved in water at pH 4.6 were compared with their wavelength shifts of the second LPG band (Figure 7c). The sensitivity of the MI film to CR (4 mM) is about 13-fold larger than that of the NI film, indicating the higher selectivity of the MI film to CR; however, this selectivity of the MI film is reduced by almost half when the CR concentration is 15 mM. This suggests that more selective CR binding to the MI film can be achieved at lower CR concentrations. Furthermore, the selectivity of the MI film to CR is estimated to be approximately 1.1, 5.5, 1.1, 2.1, and 3.9 for lysine, arginine, histidine, glutamic acid, and aspartic acid, respectively (each 4 mM).

Usually, normal human urine samples are almost neutral, and it is an essential task to investigate the pH dependence of the sensor response. For this purpose, sensitivity tests were conducted using the samples without adjusting the pH dissolved in water. Figure 8a compares the wavelength shifts of the first and second LPG bands upon exposure of the MI film-modified LPG sensor to the template and guest molecules (each 4 mM, Figure S4 and Table S1). Among the guest molecules, four amino acids—lysine, arginine, histidine, and asparagine—(Figure 8b) showed larger responses than the template and other guest molecules. Except for asparagine, they were classified as amino acids with positively charged side chains and showed weak alkalinity when dissolved in water. These positively charged amino acids are the main constituents of urine and can be considered as potential substances that interfere with CR binding to the imprint site. Therefore, the enhanced responses of the three amino acids might be attributed to their stronger electrostatic interactions with the CR BS. Probably, the $pK_{a2}$ values for their positively charged side chains were vital factors to induce their relatively high responses (Table S1). However, their higher responses could not be explained by basicity because the basicity of arginine was approximately 100 times larger than that of lysine, but the sensor response to lysine was larger than to arginine.

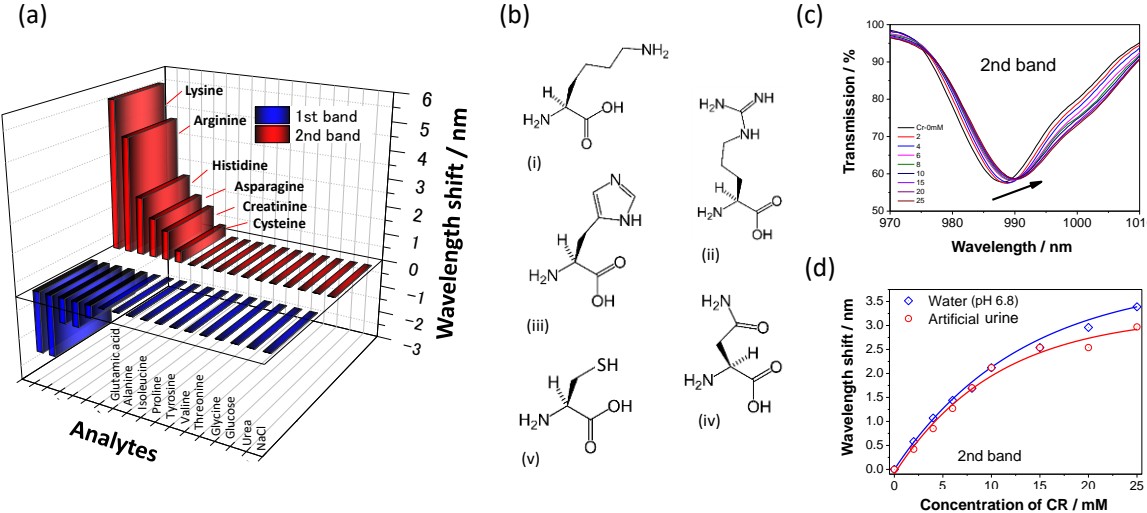

**Figure 8.** (**a**) Comparison of the wavelength shifts upon exposure of the MI film-modified LPG sensor to 4 mM of the template and guest molecules without adjusting the pH dissolved in water. (**b**) Chemical structures of five guest molecules showing relatively high responses: (**i**) lysine, (**ii**) arginine, (**iii**) histidine, (**iv**) asparagine, and (**v**) cysteine. (**c**) Changes in the TS of the second resonance band upon exposure of the MI film-modified LPG sensor to different concentrations of CR in artificial urine. (**d**) Comparison of calibration curves plotted from the wavelength shifts of the second LPG band for CR detection in water and artificial urine.

Similar selectivity tests were conducted at pH 8.0 using the guest molecules used in Figure 7c with the MI and NI films. From Figure S5, the responses of lysine, arginine, and histidine were considerably enhanced; however, the MI film retains a higher sensitivity to all analytes than the NI film. Remarkably, all analytes used for the selectivity test showed enhanced responses, whereas no significant change was observed when the pH was unadjusted. Amino acids with negatively charged side chains and neutral polar or hydrophobic side chains, glucose, urea, and NaCl were almost neutral when dissolved in water. Unless the pH was adjusted to alkaline, they did not show meaningful responses.

Moreover, asparagine with an amide group as a side chain showed notable binding compared with the other guest molecules (Figure 8a), indicating that the CR-imprinted BS allowed hydrogen bonding with guest molecules in addition to the electrostatic interaction. Despite the highest basicity of arginine, its binding to the imprinted site at pH 4.6 was suppressed than those of lysine and histidine. The diminished response of arginine also suggested that the CR-imprinted site did not match the shape of the bulky side chain of arginine.

When the guest binding was individually conducted using the samples at pH > 6.8, the responses of the positively charged amino acids were more significant than that of CR. To confirm the specificity (cross-sensitivity) of the MI film to the positively charged analytes, we conducted CR rebinding tests using artificial urine containing 11 amino acids (each of 5 mM), four urinary organic constituents, and five inorganic salts (Table 1). Figure 8c shows the responses of the MI film-modified LPG sensor to different concentrations of CR in artificial urine. The calibration curves based on the wavelength shifts of the second LPD band upon CR binding in two media of water (pH 6.8) and artificial urine (near-neutral pH) are also compared in Figure 8d. Remarkably, the presence of amino acids with positively charged side chains in artificial urine did not significantly affect the CR sensing of the MI film-modified LPG sensor. The binding constant calculated from a Benesi–Hildebrand plot for the CR template binding was estimated to be 54 $M^{-1}$ ($R^2 = 0.947$, data not shown), which corresponded to 87% and 80% of the values obtained from the wavelength shifts of the second resonance band recorded in water at pH 4.6 and 6.8, respectively (Figure 6d). This result proved that positively charged amino acids were nonspecifically bound to the imprinted site and could be easily substituted with the CR template added in the artificial urine. This unabated sensitivity to CR in the artificial urine, showing a complex matrix, indicated that strong CR binding into the CR-imprinted site created in a PAA-assisted LPD-based $TiO_2$ film was available.

Interestingly, similar CR-selective binding features are observed when an interfering substance is co-present with CR in a solution. As can be seen from Figure S6, no discernible difference was observed between the wavelength shifts upon exposure of the MI film-modified LPG sensor to CR (4 mM) and a binary mixture including CR (4 mM) and a guest molecule (5 mM), where all solutions for the selectivity test were neutralized to pH 6.8. As discussed previously, when the MI film is applied to the individual guest molecules, relatively high responses are observed for amino acids with positively charged side chains, such as lysine and histidine, capable of interacting via electrostatic interactions with the CR binding sites. Despite the presence of an interfering substance, the CR is preferentially accessible to the imprinted cavity.

### 3.6. Characteristics of the CR-Imprinted Film and CR Sensing Mechanism

XPS analysis was performed using the PAA-assisted $TiO_2$ samples with and without CR (5 and 0 mM against the 1 mM PAA in the LPD coating solution, respectively) to confirm the detailed film characteristics indirectly. Figure 9a shows survey XPS spectra for the samples of PAA/$TiO_2$ (line in red) and CR@PAA/$TiO_2$ (line in black). Remarkably, no N 1s peak was observed in the PAA/$TiO_2$ sample without CR. The high-resolution C 1s spectrum for the CR@PAA/$TiO_2$ sample consisted of five components (Figure 9b), with the most intense peak at 285.0 eV. This peak corresponded to α and β carbon atoms in the PAA and to α carbon atoms in the CR ring, which were presented as C-C and C-H

bonds [37]. The components centered at 286.6 and 287.8 eV were relevant to the bond between C and N in CR, attributable to the $=C-N^+$ and $-C=N^+$ bonds [38]. Interestingly, the peak area of the $=C-N^+$ bond was much larger than that of the $-C=N^+$ bond, suggesting that the equilibrium of CR in the film was biased toward the amine form. In addition, the C=O species identified in the CR was observed at 288.6 eV [38,39]. Finally, the component at the highest binding energy of 289.4 eV was assigned to the COOH group of the PAA.

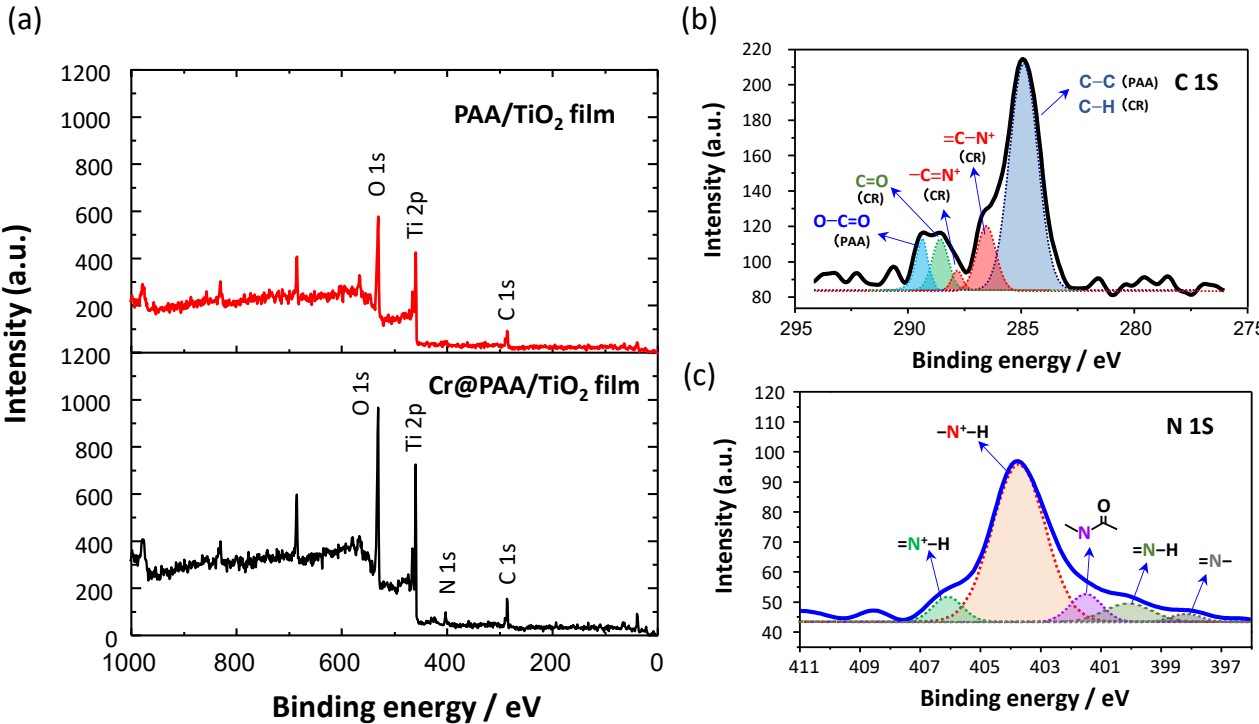

**Figure 9.** (**a**) Survey XPS spectra for samples of PAA/TiO$_2$ (line in red) and CR@PAA/TiO$_2$ (line in black). High-resolution (**b**) C 1s and (**c**) N 1s XPS peaks recorded on the CR@PAA/TiO$_2$ sample.

The high-resolution N 1s peak recorded on the CR@PAA/TiO$_2$ sample showed the presence of five components (Figure 9c), In line with the literature, the component at around 400 eV was assigned to neutral nitrogen in the CR ring (-N-H structures) [38]. The components with low binding energies, at around 401.5 and 398.0 eV, corresponded to neutral amide (-N-C=O) and imine (=N-) structures, respectively [38,40]. Notably, positively charged components, such as -N$^+$-H and =N$^+$-H species, largely shifted to higher binding energies than those reported in the literature [38]. These protonated species were bound to the surrounding TiO$_2$ matrices via electrostatic interaction, resulting in shifts to higher binding energies. Further, the -N$^+$-H bond at 403.8 eV showed a larger peak area than the =N$^+$-H species at 406.1 eV, which agreed with the result of the =C-N$^+$ and -C=N$^+$ species assigned to C 1s peak. From this perspective, we inferred that the CR template was preferentially employed in the primary amine form in the TiO$_2$ matrix.

ATR-FTIR spectra of the solid samples of pure TiO$_2$, CR@PAA/TiO$_2$, and pristine CR are depicted in Figure S7. The ATR-FTIR spectrum of pure TiO$_2$ clearly showed three bands (Figure S7a). The first very broad band at around 3200 cm$^{-1}$ corresponded to the stretching vibration of O-H groups on the TiO$_2$ surface. The second and third bands observed at around 1623 and 1414 cm$^{-1}$ corresponded to the bending mode of hydrated TiOH and the stretching vibration of the Ti-O bond, respectively [41,42]. The ATR-FTIR spectrum of the CR@PAA/TiO$_2$ sample displayed characteristic bands at 3185, 1625, and 1427 cm$^{-1}$, corresponding to the symmetric stretching of N-H bond, N-H bending, and C-H bending assigned to a CH$_2$ or CH$_3$ moiety, respectively (Figure S7b). The N-H symmetric stretching observed at 3185 cm$^{-1}$ originates the wavenumber at 3030 cm$^{-1}$ in the pristine CR sample

(Figure S7c). Similar frequency shifts were observed when the CR was complexed with metal ions [43]. In addition, the CR@PAA/TiO$_2$ sample revealed a weak shoulder assigned to the stretching of the C=O bond at 1707 cm$^{-1}$, which originated at 1702 cm$^{-1}$ in the pristine CR sample. With reference to the literature, the pure PAA showed the C=O stretching vibration (strong) at 1719 cm$^{-1}$ and the broad absorption peak at 3448 cm$^{-1}$, attributed to the OH stretching vibration [44]. However, the stretching of the free carboxylic acid (1719 cm$^{-1}$) was not found in the CR@PAA/TiO$_2$ sample. Moreover, two peaks due to titanium carboxylate are usually observed in the frequency range of 1400–1500 cm$^{-1}$ [22]. This was confirmed by the enhanced peak at around 1427 cm$^{-1}$ in the CR@PAA/TiO$_2$ sample. Plausibly, the strong band appeared as a complex of several vibration modes, such as monodentate titanium carboxylate and C-H bending assigned to the CH$_2$ and CH$_3$ moieties in PAA and CR. Further, ring-stretching modes of CR at 1242, 1207, and 1180 cm$^{-1}$ in pristine CR were observed as small peaks centered at 1191 cm$^{-1}$. Most vibration modes of the CR were confirmed in the ATR-FTIR spectrum of the CR@PAA/TiO$_2$ sample, although some peaks shifted toward high frequencies. Consequently, these ATR-FTIR results indicated that the CR template could be employed via the electrostatic interaction and hydrogen bonding with the PAA-assisted TiO$_2$ film matrix, resulting in CR molecular motion suppression.

As evident from the XPS and ATR-FTIR measurements, the primary amine form of CR as a template was preferentially introduced in the PAA/TiO$_2$ film matrix (Figure 10a). The CR template bound to TiO$_2$ matrices via electrostatic interaction could be removed through HCl (pH 3.5) washing under moderate conditions (Equation (5)). In our previous study [45], an LPD-based TiO$_2$ revealed an isoelectric point at approximately pH 4.5, and thus the TiOH moiety inside the BS returned to the anionic form again, TiO$^-$ (Equation (6)). As can be seen from the p$K_a$ values of the guest molecules (Table S1), CR was positively charged in the pH range of 5–10. Consequently, the protonated CR template could be rebound to the imprinted cavity via electrostatic interaction.

$$\text{Ti-O}^-\text{H}_3\text{N}^+\text{-CR} + \text{HCl} \leftrightarrows \text{Ti-OH} + \text{CR-N}^+\text{H}_3\text{Cl}^- \tag{5}$$

$$\text{TiOH} + \text{H}_2\text{O} \leftrightarrows \text{TiO}^- + \text{H}_3\text{O}^+ \tag{6}$$

The PDDA/SiO$_2$ NPs base coating played a crucial role in improving the sensitivity of the LPG sensor because the low RI-SiO$_2$ NPs could yield a high surface for efficient LPD-based CR imprinting. In addition, SiO$_2$ NPs assembled in the base coating acted as scavengers of HF produced by the hydrolysis of TiF$_6{}^{2-}$ ions, which helped accelerate the LPD process. This was confirmed from an SEM image in Figure 10b, wherein SiO$_2$ NPs were shrunk, showing different surface morphology compared with that of the same film before LPD-based TiO$_2$ coating (Figure 2a). However, the film thickness was unchangeable, even after LPD-based TiO$_2$ coating, showing ca. 330 nm in thickness, similar to that of the original base coating. This morphological observation suggests that SiO$_2$ NPs are networked with TiO$_2$ via HF etching during the LPD process.

Finally, the optimal condition for CR imprinting could be determined by controlling the PAA concentration employed in the TiO$_2$ film matrix. As expected from the QCM and LPG experiments, the most essential role of PAA was to suppress the high activity of TiO$_2$ to minimize the nonspecific adsorption of guest molecules. For low PAA contents or without PAA, the film growth was relatively fast. Thus, morphologically uncontrolled particles were formed (Figure 10c), as reported in our previous study [21], which made efficient binding sites that reflected the structure of the template molecule difficult. Meanwhile, high PAA contents inhibited TiO$_2$ film growth because the complexation of TiF$_6{}^{2-}$ precursors with PAA in the film-forming disturbed the hydrolysis by water for sol-gel reaction. Very high PAA content significantly reduced the activity of TiO$_2$, resulting in insufficient reactivity with CR. As a result, binding sites for CR could not be sufficiently formed in the TiO$_2$ film. In this study, approximately 1 mM of PAA as unit molar concentration was determined as optimal concentration for CR imprinting. We inferred that PAA can be considered an

additive to control the film morphology and TiO$_2$ activity rather than to provide binding sites based on the acid–base reaction with templated molecules.

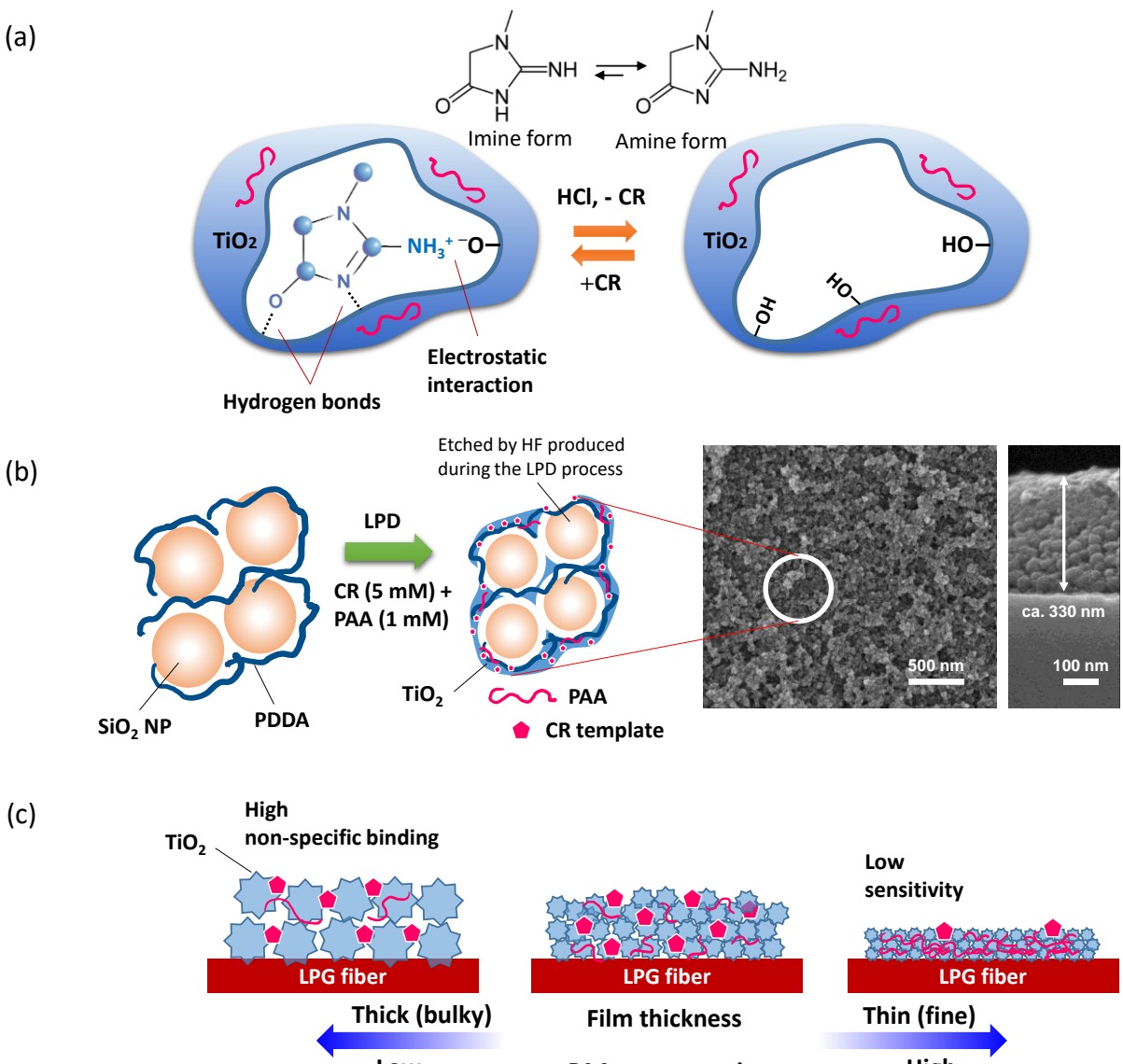

**Figure 10.** (**a**) Schematic of the CR-imprinted site in the PAA-assisted TiO$_2$ film matrix and available interactions inside the binding cavity. (**b**) Role of the PDDA/SiO$_2$ NPs base coating in improving the sensitivity of the LPG, where SiO$_2$ NPs are etched by HF produced by the hydrolysis of TiF$_6$$^{2-}$ ions, resulting in the acceleration of LPD process. SEM images show the surface morphology and cross-section of the base coating of a (PDDA/SiO$_2$ NPs)$_{10}$ film after LPD-based TiO$_2$ coating on a 9 MHz QCM electrode. (**c**) Dependence of the PAA content on TiO$_2$ film growth and morphology control.

　　　　Several optical fiber sensors for CR detection were recently reported. For example, Sharma et al. have proposed an optical fiber sensor based on lossy mode resonance modified with a molecularly imprinted organic polymer (methacrylic acid) film for CR detection with a LOD of 0.02 mM [46]. Menon et al. proposed a surface plasmon resonance (SPR) CR sensor based on Kretschmann configuration employing enzymatic reaction; however, no selective response was reported for this sensor [47]. In the current work, we proposed a novel approach for sensitive and selective CR detection using molecularly imprinted TiO$_2$ inorganic matrices coupled with fiber-optic LPGs. More details of the conventional optical sensors for CR detection are compared with the current work in Table 3.

**Table 3.** A comparison of the analytical performance with other CR optical detection methods previously reported in the literature.

| Sensor | Coating | Sensitivity (LOD) | Range | Response Time | Selectivity (Cross-Sensitivity) | Reference |
|---|---|---|---|---|---|---|
| Lossy mode resonance | Methacrylic acid–base molecularly imprinted polymer | 0.41 nm/$(\mu g\ mL^{-1})$ ($1.86\ \mu g\ mL^{-1}$ $\approx 0.02$ mM) | 0–16 mM | <1 min | Dopamine, urea, ascorbic acid | [46] |
| SPR | Creatininase enzyme | $4°$/M (−) | 10–200 mM | – | – | [47] |
| SPR | Molecularly imprinted polymer | $18.81 \times 10^{-2}$ nm/$\mu$M (1.17 $\mu$M) | | – | – | [48] |
| Photonic crystal | – | − (−) | 0.080–0.086 mM | – | – | [49] |
| This work | LPD-based $TiO_2$ | 0.2 nm/mM (0.01 mM) | 0–25 mM | <1 min | Glucose, spermine, sarcosine, caffeine, urea, uric acid, 14 *L*-amino acids | |

## 4. Conclusions

A novel imprinting method using LPD-based $TiO_2$ nanothin films was proposed for sensitive and selective CR detection on fiber optic LPGs. The incorporation of PAA in the film helped control the film morphology and $TiO_2$ activity inside the binding site. An optimal CR imprinting could be achieved with a 5:1 ratio of CR to PAA in the LPD coating solution. The MI film deposited on LPGs exhibited the highest sensitivity to CR dissolved in water at pH 4.6, whereas the NI film represented small changes under the same conditions. The optimal pH for CR binding was considered to be 4.6, at which both films revealed a nearly ninefold difference in their binding constants to CR. Moreover, the CR selectivity of the MI film decreased as the nonspecific binding to the NI film increased with the increase in pH. Usually, normal human urine samples are almost neutral, and it will be meaningful to develop a pH-independent sensor. Fortunately, the pH application range of the current LPG sensor was available up to neutral, and there was no significant decrease in sensitivity at near-neutral pH. For instance, the wavelength shifts of the second LPD band upon CR binding were almost the same, even in the artificial urine with a complex matrix. The current LPG sensor enables label-free CR sensing in the tens of mM concentration range, which covers an 8.8–15.9 mM average concentration range of urinary CR [50]. The proposed method provides a useful methodology for creating sensing devices coupled with an LPD-based molecular imprinting technique, particularly for biological sensing. In addition, the current approach is simple and easy to extend into miniature devices, and the durability and biocompatibility of $TiO_2$ MI films to chemicals can offer a cheap and label-free way to clinical systems. Our future work is to improve the proposed sensor system by searching for pH-resistive materials as additives in LPD-based imprinted films for real urine sample analysis.

**Supplementary Materials:** The following are available online at https://www.mdpi.com/article/10.3390/chemosensors9070185/s1, Figure S1: 9 MHz QCM frequency shifts measured during the same LbL electrostatic self-assembly of PDDA and $SiO_2$ NPs, Figure S2: Changes in the TS of the LPG with a base coating during the NI-film deposition, Figure S3: Changes in the TS of the LPG without a base coating during the CR@PAA/$TiO_2$ (2 mM CR and 6 mM PAA) film deposition, Figure S4: Chemical structures of amino acid guest molecules used in this study, Figure S5: Comparison of the wavelength shifts of the second LPD band upon exposure of the NI and MI films to the template and guest molecules, Figure S6: Comparison of the wavelength shifts of the second LPG band upon exposure of the MI film to CR (4 mM) or a binary mixture including CR (4 mM) at pH 6.8, Figure S7: ATR-FTIR spectra of the solid samples of pure $TiO_2$, CR@PAA/$TiO_2$, and pristine CR, Table S1: Molecular properties of guest molecules including amino acids used in this study.

**Author Contributions:** Conceptualization, S.-W.L.; methodology, S.-W.L. and S.A.; validation, S.A.; formal analysis, S.A. and T.W.; investigation, S.A. and T.W.; data curation, S.A., T.W., Y.P., S.M. (Sota Matsuzaki) and S.T.; writing—original draft preparation, S.-W.L. and S.A.; writing—review and editing, S.-W.L., S.K. and S.J.; supervision, S.-W.L.; project administration, S.-W.L.; funding acquisition, S.-W.L. and S.M. (Shigekiyo Matsumoto). All authors have read and agreed to the published version of the manuscript.

**Funding:** This work was funded by the grant-in-aid for scientific research (C) (20K09223) from the Japan Society for the Promotion of Science.

**Institutional Review Board Statement:** Not applicable.

**Informed Consent Statement:** Not applicable.

**Data Availability Statement:** Not applicable.

**Conflicts of Interest:** The authors declare no conflict of interest.

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
