# Peer review of "Label-Free Creatinine Optical Sensing Using Molecularly Imprinted Titanium Dioxide-Polycarboxylic Acid Hybrid Thin Films: A Preliminary Study for Urine Sample Analysis"

_chemosensors, doi:10.3390/chemosensors9070185_

Round 1

Reviewer 1 Report

Please find attached my comments

Reviewer 2 Report

The paper reports on label-free detection of crating by a LPG modified with MI TiO2 PAA film. All the chemical functionalizations and the characterization procedures of the materials and devices are well described. The main concern is on results presentation. All the quantities measured or estimated in the text (sensitivity, limit of detection and so on) must be reported together with their errors otherwise they are meaningless. Errors must be written with only one significant digit so that, for example, frequencies on page 7 are 28.4 ± 16.8 and 1430 ± 115 Hz should be  30± 17 and 1430 ± 110 Hz and so on. Authors claim about selectivity but from the data reported they only proved specificity. According to IUPAC definitions, specificity is to give a different instrumental response to different analyte (and it does not imply selectivity), while selectivity is the ability to quantify a single analyte in a complex matrix (and it implies specificity). Please, add experimental proof of selectivity or correct the text changing selectivity in specificity.

Reviewer 3 Report

In this manuscript, PAA-assisted molecularly imprinted TiO2 nanothin films fabricated via liquid phase deposition (LPD) were employed for CR detection. The manuscript has carried out many performance analyses of the proposed method, but the innovation is not emphasized enough, such as in Abstract, etc., need to be optimized. Some points should be clearly addressed and the manuscript should be substantially satisfactorily revised before its possible acceptance.

  1. In Figure 6d, how were the pH values of 6.8 and 4.6 selected, and what pH buffer was used?
  2. Why are Figure 7c and Figure S5 duplicated? The author needs to check. Since there is a very obvious response to four amino acids, indicating that the selectivity of the imprinted polymer is not good, how will it prove that its accuracy in the actual sample detection is not affected? It is also recommended to increase the detection of CR in real urine samples.
  3. The comparison with the currently published CR optical detection methods needs to be presented to reflect the advantages of this work.
  4. Error bars need to be added to the figures.
  5. Note that "CR" is capitalized in the writing of “Cr@PAA/TiO2 films”. In addition, writing like “6.70 × 101” and “1-mM PAA” are not standardized.
  6. About the references, some of the them are old and more related publications on this topic can be referenced for further improving, such as

Molecular Imprinting: Green Perspectives and Strategies, ADVANCED MATERIALS    Article Number: 2100543, Early Access: JUN 2021

Long-period grating fiber-optic sensors exploiting molecularly imprinted ……, MICROCHIMICA ACTA 2020 Volume:187, Issue:12, Article Number: 663;

L-phenylalanine-imprinted polydopamine-coated CdS/CdSe ……, BIOSENSORS & BIOELECTRONICS 2020 Volume:165, Article Number:12346;

A visible-light-driven photoelectrochemical molecularly imprinted sensor ……, ANALYST 2019 Volume:144, Issue:10, Pages:3405-3413; ……

Round 2

Reviewer 2 Report

The revised paper can be published

Reviewer 3 Report

The authors have satisfactorily responded to all the questions and made the necessary changes to the manuscript. I have no further questions and suggest the acceptance of the revised manuscript.